# Numerical Study of Knocking Combustion in a Heavy-Duty Engine under Plateau Conditions

**Haiying Li \*, Xiaoqin Zhang †, Chaofan Li † , Rulou Cao †, Weiqing Zhu †, Yaozong Li †, Fengchun Liu †
and Yufeng Li †**

China North Engine Research Institute, Tianjin 300400, China; xiaoqinshanxi@126.com (X.Z.);
lichaofan93@163.com (C.L.); 18203212302@163.com (R.C.); ck2828303@163.com (W.Z.); lyzbest@163.com (Y.L.);
15835382882@139.com (F.L.); yufeng.li@hotmail.com (Y.L.)
* Correspondence: ying_h_a@hotmail.com
† These authors contributed equally to this work.

**Abstract:** Diesel engine combustion becomes very rough and can lead even lead to deflagration under high altitude conditions, which is harmful to component durability. In this study, the effects of altitude on the main combustion characteristics—in-cylinder fluid flow, spray behavior, and pressure and temperature distribution—were analyzed with CFD. A numerical model was built on the CONVERGE platform and validated with the optical spray behavior and pressure trace measured by the test bench. The simulation results indicated that the decreases in compression pressure and temperature at 4.5 km led to an over 4 °CA longer ignition delay than those of 1 and 3 km. The combustion efficiency decreased from 90% to 47% when the combustion changed from normal combustion to knocking combustion due to severe spray impingement. The processes of end-gas ignition, sequential combustion, and pressure oscillation in knocking combustion were revealed by the numerical modeling results. These results indicate that super-knocking combustion exists in both spark-ignition (SI) engines and compression-ignition (CI) engines.

**Keywords:** knock; CFD; diesel engine; combustion; altitude

## 1. Introduction

It is known that with increases in altitude, air density and oxygen content decline. Few investigations of combustion characteristics and performance for ground vehicles at elevated altitudes have been conducted recent years, though atmospheric conditions below 2000 m have almost no effects on combustion and engine performance. However, the deterioration of engine performance (including increases in fuel consumption rate, decreases in brake thermal efficiency (BTE), and the extension of ignition delay_ occurs when engines operate at an altitude above 4000 m [1,2]. Wang et al. [2] found that brake thermal efficiency decreased by about 20% as altitude increased from 3.3 to 4.5 km with a moderate load at a speed below 1200 rpm in a 16.9 L, heavy-duty, turbocharged, common-rail diesel engine on a mobile test bench. Focusing on the operation reliability of engines at high altitudes, various experiments regarding the effects of altitude on combustion characteristics and engine performance have been conducted in the China North Engine Research Institute (CNERI). The effects of fuel properties on combustion characteristics and cycle-to-cycle variation (COV) were studied in single cylinder diesel engines by Cai et al. [3] and Zhang et al. [4]. Furthermore, Kan et al. [5] found that the COV of the IMEP increased by 0.6% for every 1 km rise in altitude, resulting in mixture heterogeneity at high altitudes, with a 2.8 L, V6, heavy-duty diesel engine. Li et al. [6] observed knocking combustion with less than 20% BTE in this engine when it operated with a partial load, at 1200 rpm, and when atmospheric pressure was reduced to 57.6 kPa in response to the pressure conditions of 4.5 km. Additionally, the mean Peak Pressure Rise Rate increased from 24 to 56 bar/°CA when simulated altitude conditions changed from 1 to 4.5 km, and a correlation between

knock intensity (KI) and Peak Pressure Rise Rate was obtained. Knocking is well-known to result from the autoignition of end-gas-associated severe pressure oscillations [7], and it resembles knock in spark-ignition (SI) engines. Compared to SI knock or super-knock, CI knocking combustion can be relatively clearly identified from pressure oscillation or knock intensity. Li et al. [6] firstly discovered the existence of knocking combustion in a V6, heavy-duty, CI diesel engine using by a plateau simulation test bench under the altitude condition of 4.5 km. As illustrated in Figure 1, although the pressure rise of CI knocking combustion reprinted from the experimental pressure data in [6] was lower than that of SI super-knock, it was still in the order of that of conventional SI knock.

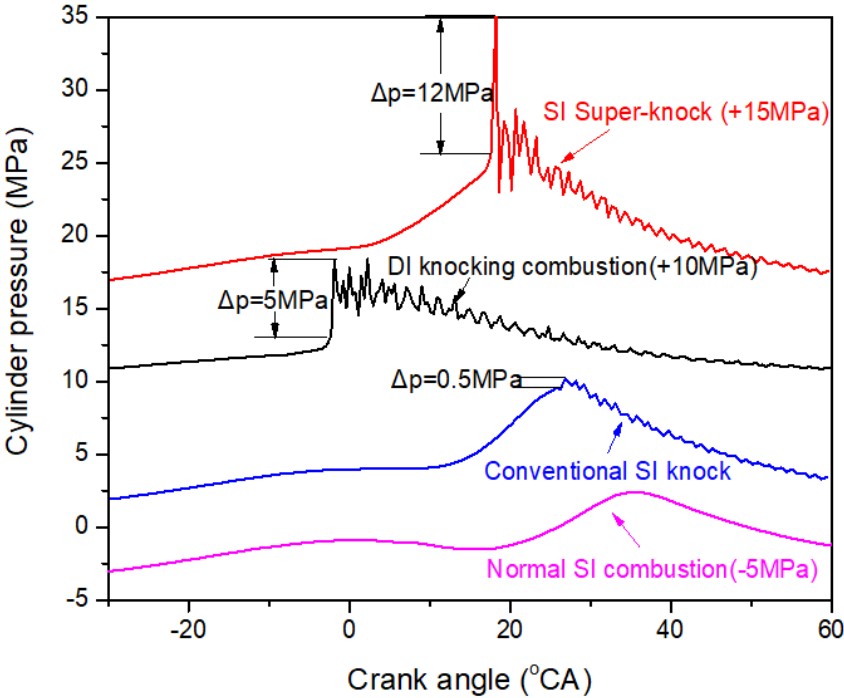

**Figure 1.** Three typical modes of combustion in boost SI engines (reprinted from [8]) and CI knocking combustion pressures traces.

The combustion pressure waves resulting in this pressure oscillation are firstly generated by local explosions that occur at places with local higher fuel concentrations and/or temperatures. Then local pressure and temperature near the combustion wave front can be increased by the compression of pressure fluctuation when the pressure waves transmit across the remaining unburned gas [9]. In particular, when a local explosion occurs near the cylinder wall, the pressure fluctuation is enhanced by wave reflection on the wall [10]. By exploring the interaction of the autoignition flames and reflection waves of pressure on the wall, Terashima et al. [11] discovered that the intensity of pressure oscillation was strongly related to the position of autoignition and the size of kernels. An unburned zone with the most reactive composition and the highest temperature is inclined to result in autoignition. In other words, the thermal and compositional stratification of unburned gas is crucial to knock onset and flame propagation. Although the experimental results of a simulation plateau bench indicated the space-averaged combustion characteristics of knocking combustion in a CI engine [6], the position of ignition and the style of flame propagation cannot be measured with a bench test.

With consideration of the stochastic behavior of knocking combustion, Computation Fluid Dynamics (CFD), especially Large Eddy Simulation (LES), has become necessary for the study of knocking combustion [12]. The effects of local thermal and compositional stratification on local autoignition and engine knock have been researched in several numerical studies [13–15].

However, no LES studies have focused on the formation of CI knocking combustion under practical engine conditions. In this study, LES simulations of engine conditions at various altitudes in bench test were carried out in order to obtain a deeper understanding of knocking combustion. Firstly, the effects of altitude on the main combustion characteristics—spray development, fuel–air mixing, and temperature distribution—were investigated. Secondly, local thermal and chemical information such as pressure, temperature, and concentration were calculated with the LES technique during the knocking combustion process, especially during knock onset. Thus, the position of ignition, pressure wave, and flame propagation of CI knocking combustion were revealed.

## 2. Experimental Methods

The bench test was carried out in a V6, heavy-duty, intercooled-turbocharged diesel engine with a displacement of 16.92 L. The injected fuel was No.35 diesel. An Imtech$^{TM}$ plateau test system—which consisted of an air filter, inlet fan, inlet drying device, frequency conversion fan, inlet temperature regulator, inlet humidity regulator, refrigeration compressor, steam generator, steam heater, inlet humidity control system, exhaust cooling, exhaust fan, cooling water system, and backwater system—was used in this study for the simulation of various atmospheric pressures at different altitude conditions. The details of this test bench were described in [6].

The instantaneous in-cylinder pressure was measured with a water-cooled piezoelectric crystal sensor Kistler$^{TM}$ 6061 B, with a natural frequency of 90 kHz, that enabled the detection of knock.

## 3. Numerical Methodology

Three-dimensional (3D) fluid flow and combustion modeling were performed with CONVERGETM software. In order to reproduce ignition, premixed combustion, and diffusion combustion, a SAGE detailed chemistry solver [16] was adopted. The transport equations were solved by SAGE to calculate the reaction rates of each elementary reaction.

The four-species (n-decane, iso-octane, methylcyclohexane, and toluene) skeletal oxidation mechanism [17] was selected to assess the combustion process. This mechanism consists of 70 species and 220 reactions with verified predictability in a wide temperature range, as well as for practical diesel fuel under wide operating conditions.

In order to solve a sub-filter-scale (SFS) kinetic energy transport equation, a one-equation sub-grid scale (SGS) eddy viscosity model [16], in which SFS turbulent kinetic energy is solved by a turbulent viscosity model, was used. This model has been widely adopted by researchers to reproduce complex flows with the adaption of varied filter sizes.

## 4. Numerical Model Setup

Three-dimensional (3D) fluid flow and combustion modeling was performed with CONVERGETM software, with turbulence, evaporation, droplet breakup, spray–wall interaction, particle interaction, collision, and combustion submodels, as presented in Table 1. The activation of adaptive mesh refinement in terms of velocity and temperature with scale of 3 and a calculation time-step refinement (minimum time-step = $1 \times 10^{-9}$ s; maximum time-step = $1 \times 10^{-6}$ s) was undertaken in order to capture the knocking combustion process. This software uses a cut-cell technology and forced orthogonal gridding with Adaptive Mesh Refinement (AMR) and grid embedding to fully generate the mesh instead of requiring extensive user input [16], and it has a base grid size of 2 mm, which is beneficial to cut-off the time for preprocessing, especially for parallel computing. Thus, the total mesh number at TDC was 4,236,757 cells, and the maximum was 4,445,155 cells. Figure 2 illustrates the geometry, including the structure of valve pockets and the computational mesh at TDC.

**Table 1.** Submodels used in this study.

| Submodel | Name |
|---|---|
| Turbulence | LES |
| Evaporation | Frossling |
| Droplet breakup | KH |
| Spray–wall interaction | Bai–Gosman |
| Collision | NTC |
| Combustion | Four species skeletal oxidation mechanism [17] |

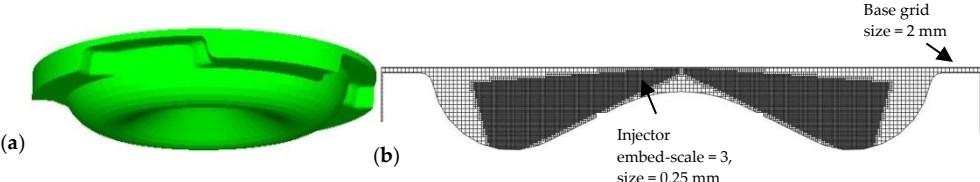

**Figure 2.** Illustration of geometry and the computational mesh at TDC: (**a**) Geometry of engine for modelling; (**b**) illustration of computational mesh at TDC. (Amr-velocity (1 m/s), scale = 3, and size = 0.25 mm; Amr-temperature (2.5 K), scale = 3, and size = 0.25 mm).

Numerical simulation was undertaken from IVC to EVO, that is, from −126 to 100 °CA ATDC. Initial computational conditions were provided by one-dimensional (1D) model results, as listed in Table 2. The fuel injection rate is shown in Figure 3.

**Table 2.** Initial and boundary conditions of simulation.

| Altitude (km) | 4.5 | 3 | 1 |
|---|---|---|---|
| Mean pressure @IVC (kPa) | 85 | 154 | 180 |
| Mean temperature | 402 | 422 | 422 |
| @IVC (K) | 550 | 550 | 550 |
| Piston temperature (K) | 450 | 450 | 450 |
| Liner temperature (K) Head temperature (K) | 500 | 500 | 500 |
| Injection quantity per cycle (mg) | 225 | 225 | 225 |

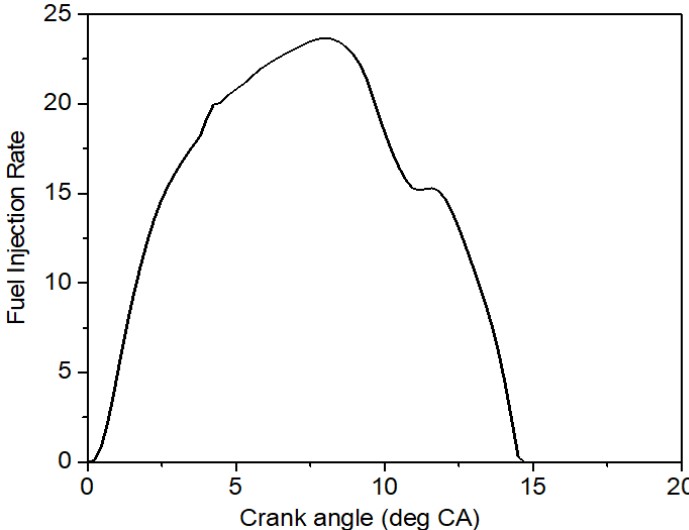

**Figure 3.** Profile of fuel injection rate.

## 5. Validation of Numerical Modeling

The spray breakup model was calibrated with the spray behavior captured with the high-speed camera in the constant volume vessel. For example, Figure 4 illustrates a comparison of experimental and numerical image of spray under the conditions of an injection pulse of 2 ms, an injection pressure of 40 MPa, an ambient pressure of 3.5 MPa, and an ambient temperature of 300 K. As shown in Figure 4, the shape and penetration of spray calculated by CFD basically agreed with those captured by the high-speed photos. Additionally, the predicted vapor penetration length was in good agreement with measurements under a high temperature of 600 K, as shown in Figure 5.

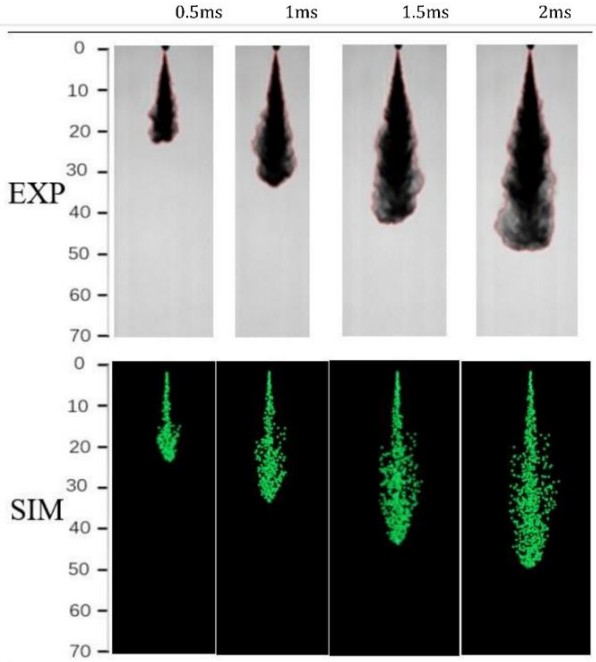

**Figure 4.** Comparison of experimental and numerical images of spray behavior (injection pulse = 2 ms, injection pressure = 40 MPa, ambient pressure = 3.5 MPa, and ambient temperature = 300 K).

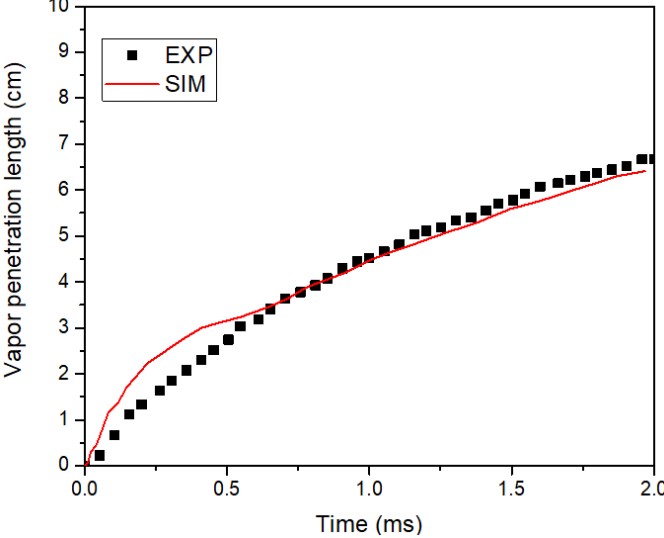

**Figure 5.** Comparison of experimental and numerical vapor penetration length (injection pulse = 2 ms, injection pressure = 40 MPa, ambient pressure = 3.5 MPa, and ambient temperature = 600 K).

The combustion model was calibrated with the experimental pressure trace at an altitude of 4.5 km. As shown in Figure 6, a noticeable difference between the numerical ensemble-averaged pressure and sensor-measured pressure was observed at the 4.5 km altitude, which was caused by the pressure inhomogeneity resulting from strong pressure oscillation. However, the local pressure predicted by CFD in the cell of the location near the pressure sensor was in good agreement with measurements. Additionally, consistency was observed at altitudes of 3 and 1 km, and the differences between experimental and numerical results were generally within a 2% error bar, as shown in Figures 7 and 8.

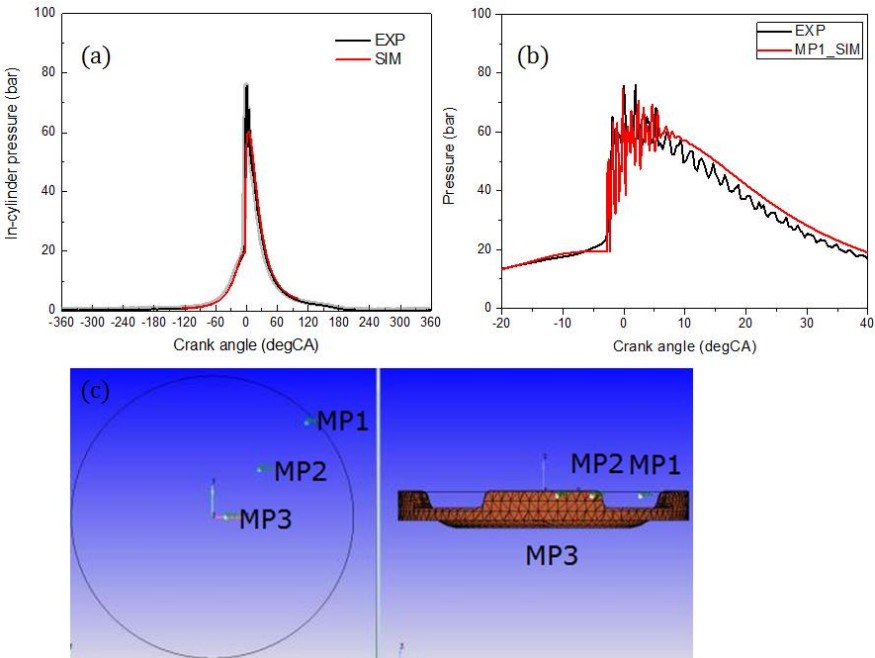

**Figure 6.** Comparison of numerical and experimental pressure at an altitude of 4.5 km: (**a**) comparison of ensemble-averaged numerical pressure and sensor-measured pressure with an error bar of 2%; (**b**) comparison of numerical and measured pressure at a location near the pressure sensor mounting; (**c**) illustration of monitor points (MP1—near the pressure sensor, which is near the liner; MP2—1/2 bore; and MP3—center of the cylinder).

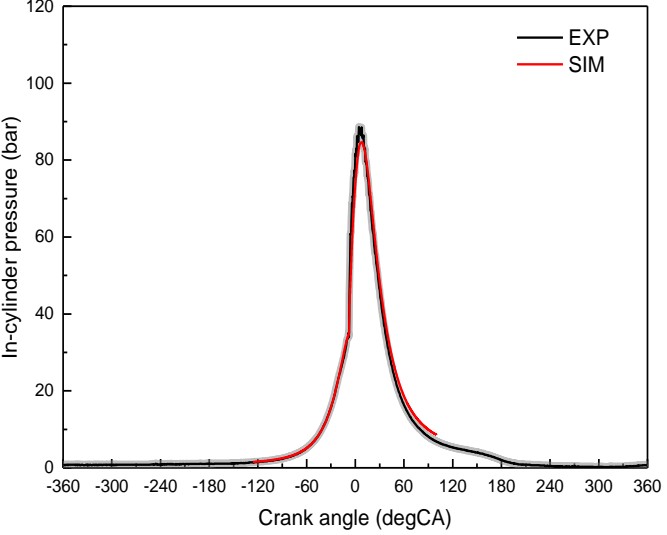

**Figure 7.** Comparison of numerical and experimental pressure with an error bar of 2% at an altitude of 3 km.

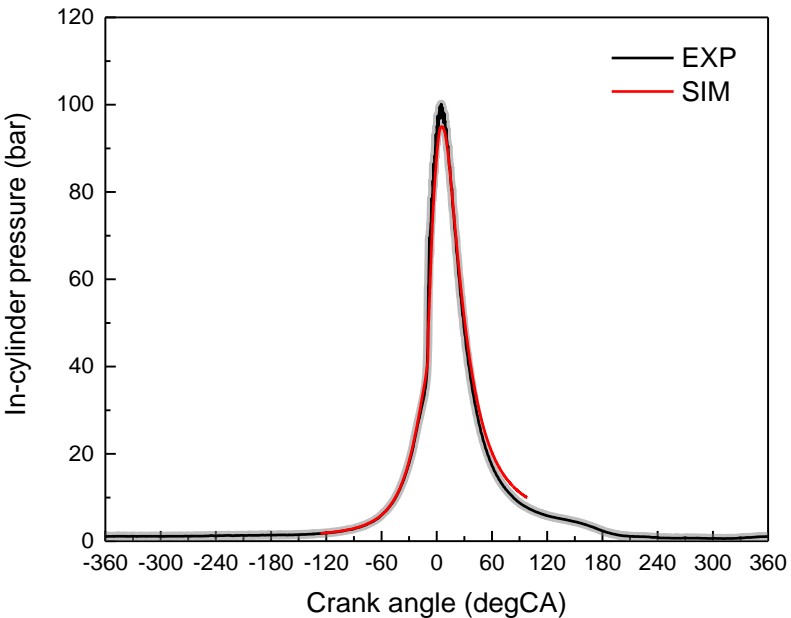

**Figure 8.** Comparison of numerical and experimental pressure with an error bar of 2% at an altitude of 1 km.

## 6. Numerical Results and Discussions

Comparisons of the mean in-cylinder pressure, temperature, and integrated heat release at various altitudes are illustrated in Figures 9–11. Table 3 shows main combustion parameters at various altitudes. In Table 3, one can observe almost no difference between the altitudes of 1 and 3 km and a significant difference between the aforementioned altitudes and the altitude of 4.5 km. In detail, with the altitude increase from 3 to 4.5 km, the maximum mean in-cylinder pressure was reached 3.3 °CA earlier and reduced by 28%, and a sharp pressure growth was observed at the altitude of 4.5 km, that is, the Peak Pressure Rise Rate (PPRR) was reached 4.8 °CA earlier and was 1.5 times higher than that of 3 km, as shown in Table 3.

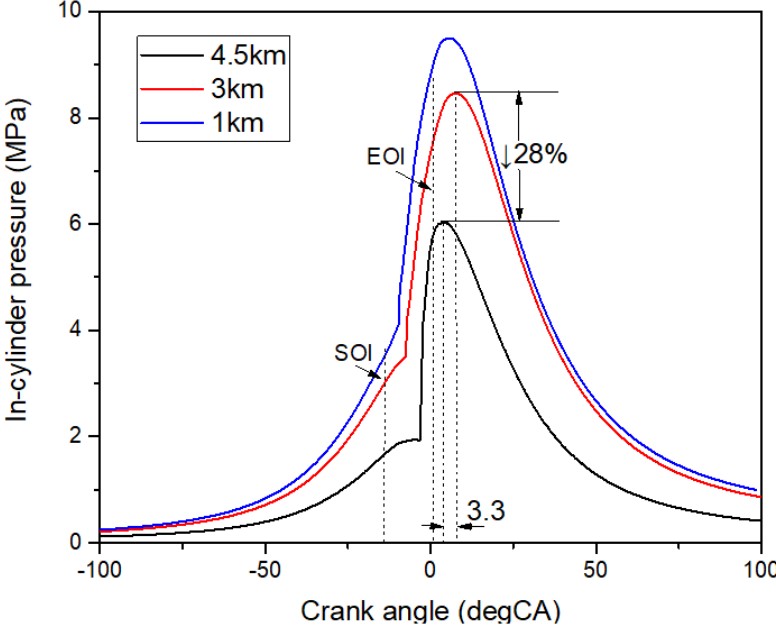

**Figure 9.** Comparison of mean in-cylinder pressure at various altitudes.

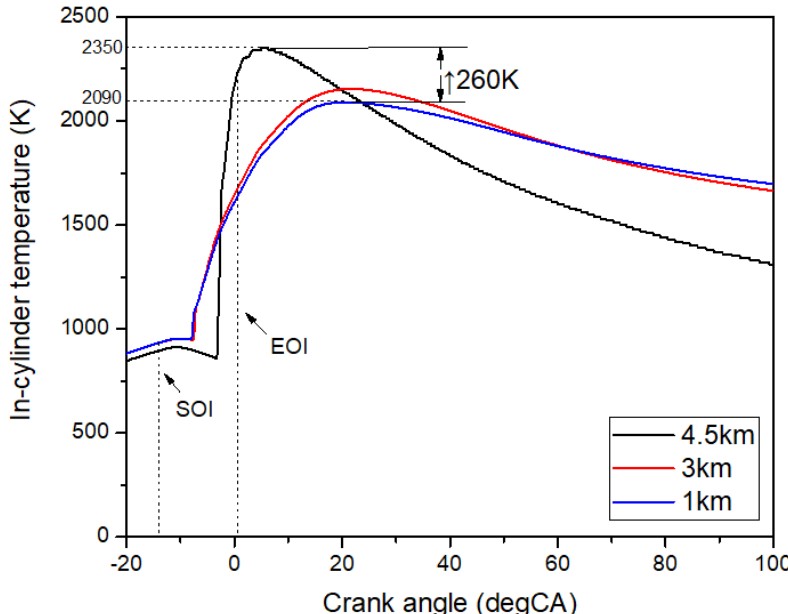

**Figure 10.** Comparison of mean in-cylinder temperature.

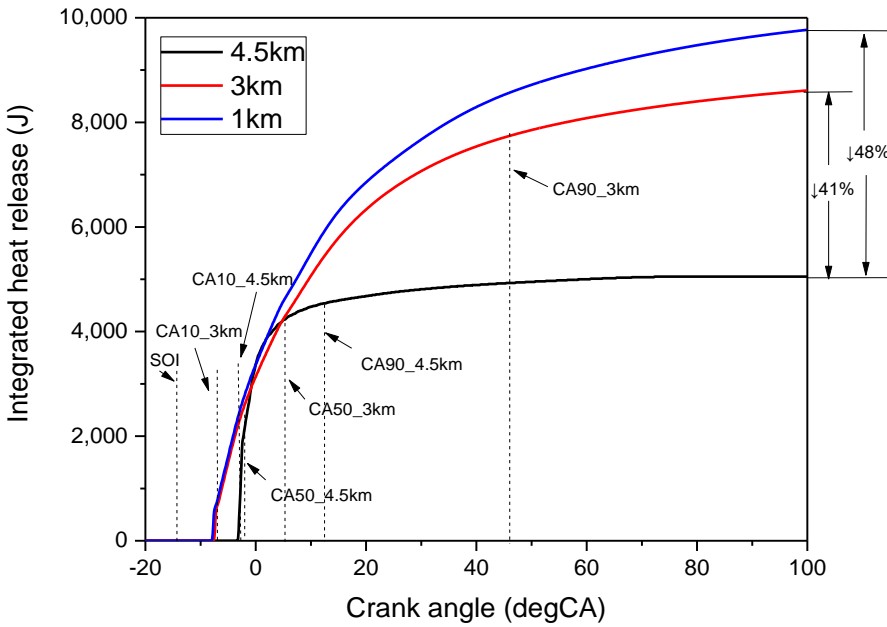

**Figure 11.** Comparison of integrated heat release at various altitudes.

**Table 3.** Effects of altitude on combustion characteristic parameters.

| Altitude (km) | 4.5 | 3 | 1 |
|---|---|---|---|
| Maximum combustion pressure (MPa) | 6.1 | 8.5 | 9.5 |
| Related crank angle of maximum combustion pressure (°CA) | 4.2 | 7.5 | 7.5 |
| PPRR (MPa/°CA) | 3.5 | 2.4 | 0.9 |
| Related crank angle of PPRR (°CA) | −2.5 | −7.3 | −7.8 |
| Maximum combustion temperature (K) | 2350 | 2154 | 2090 |
| Total mass of droplets hitting the wall (mg) | 136 | 14.5 | 7.2 |
| Maximum of heat release rate (J/°CA) | 3040 | 2078 | 1936 |

**Table 3.** *Cont.*

| Altitude (km) | 4.5 | 3 | 1 |
|---|---|---|---|
| Crank angle of maximum of heat release rate (°CA) | −2.5 | −7.3 | −7.7 |
| Integrated heat release (J) | 5050 | 8610 | 9769 |
| CA10 (°CA ATDC) | −3.1 | −7.3 | −7.7 |
| Combustion efficiency | 47% | 81% | 91% |
| CA50 (°CA ATDC) | −1.5 | 5.3 | 6.7 |
| CA90 (°CA ATDC) | 12.7 | 46.2 | 52.2 |
| Ignition delay (°CA) | 10.9 | 6.7 | 6.3 |
| Combustion duration (°CA) | 15.8 | 53.5 | 59.9 |

As shown in Figure 10, the maximum combustion temperature increased from 2090 K at 1 km to 2350 K at 4.5 km. Although the maximum pressure and the integrated heat release decreased from 1 to 4.5 km, the maximum of heat release rate increased by 57%, as listed Table 3. This indicates the occurrence of rapid and short heat release at 4.5 km. Thus, the maximum combustion temperature increased. As seen in Figure 11 and Table 3, the ignition delay of 4.5 km was 4.2 °CA longer than that of 3 km; within the duration of ignition delay, considerable ignitable fuel vapor is formed. The maximum heat release rate of 4.5 km was reached 5 °CA later and was 1.5 times higher than that of 3 km, and the integrated heat release was reduced by 41% and 48% compared to those of 3 and 1 km, respectively; thus, the combustion efficiency (defined by the ratio of integrated heat release to the product of fuel injection quantity and low heat value) of 4.5 km was only 47%, the combustion duration was only 15.8 °CA, and CA50 was reached before TDC. At 4.5 km, the main combustion duration from CA10 to CA90 was 15.8 °CA. The duration from CA10 to CA50 was 1.6 °CA, indicating rapid premixed combustion. Finally, with the long ignition delay, rapid premixed combustion and low combustion efficiency were discovered at 4.5 km.

Figures 12 and 13 show the pressure and flow velocity distribution at various altitudes, respectively. Firstly, in the cases of 1 and 3 km, combustion started around −7 °CA ATDC and the main combustion durations from CA10 to CA90 were 59.9 and 53.5 °CA, respectively. The durations from CA10 to CA50 were 14.4, 12.6, and 1.6 °CA. This was typical diffusion combustion. A random distribution of high pressure was observed in the cases of altitudes below 3 km.

In the case of 4.5 km, due to a large amount of ignitable mixture during the long ignition delay, a high-pressure zone was firstly formed at −3.2 °CA ATDC at the left side, as shown in Figure 12. This pressure convergence was possibly caused by the thermal expansion of end-gas ignition. Then, pressure propagation began at supersonic flow with a Mach number of 2–4, as shown in Figure 13. A reflection wave was formed when a pressure wave arrived at the other side of the cylinder at −2.4 °CA ATDC, so a typical reciprocating pressure oscillation was observed. In other words, the pressure wave propagated from the one side to the other side of the cylinder within 0.8 °CA and the mean propagation velocity of pressure wave was 1364 m/s, which was roughly estimated as the rate of bore divided by the duration.

Comparisons of local pressures and the power spectral density of three monitor points (MPs) near the pressure sensor (MP1), at the half of bore (MP2), and in the center of the cylinder (MP3) are shown in Figure 14 to enable the assessment of the pressure wave propagation source. Firstly, to varying degrees, local pressures at all three monitor points reproduced the characteristics of strong pressure oscillation. Furthermore, the frequencies of the first two peaks (3916 and 5626 Hz) at the MP1 and MP2 locations were similar to the theoretical values of the first and second oscillation modes (modes 1 and 2), as discussed in [6], and the frequency (5626 Hz) of the first peak at the MP3 location was similar to the

theoretical value of the second oscillation mode (mode 2). These data are consistent with the pressure propagation shown in Figure 12. That is, the pressure wave of mode 1 started near the wall due to local ignition and moved towards the center of the cylinder, mode 2 was formed when a pressure wave arrived at the center of the cylinder, and then other modes emerged. Figure 15 shows the spray behavior at various altitudes. Compared to the cases of altitudes of 1 and 3 km, more droplet particles could be found at the altitude of 4.5 km due to bad evaporation. Additionally, with the increase in altitude from 1 to 4.5 km, spray tip penetration extended from 50 to 70 mm, as shown in Figure 16. Longer spray tip penetration led to severe wall wetting. Noticeably, the total mass of droplets hitting the wall at 4.5 km was 9.4 times larger than that at 3 km, which was 53% of the total mass of fuel injection (as listed in Table 3). A large amount of wall film was formed, and the evaporation of liquid fuel changed from small droplets to wall film evaporation. Thus, both decreases in the evaporation rate and increases in the absorption of heat during evaporation occurred. The in-cylinder temperature decreased after the SOI, as shown in Figure 14. Almost 30% of injected fuel remained on the wall at 20 °CA ATDC in the case of 4.5 km, while less than 4% remained in the 152 cases of altitudes below 3 km, as seen in Figure 17.

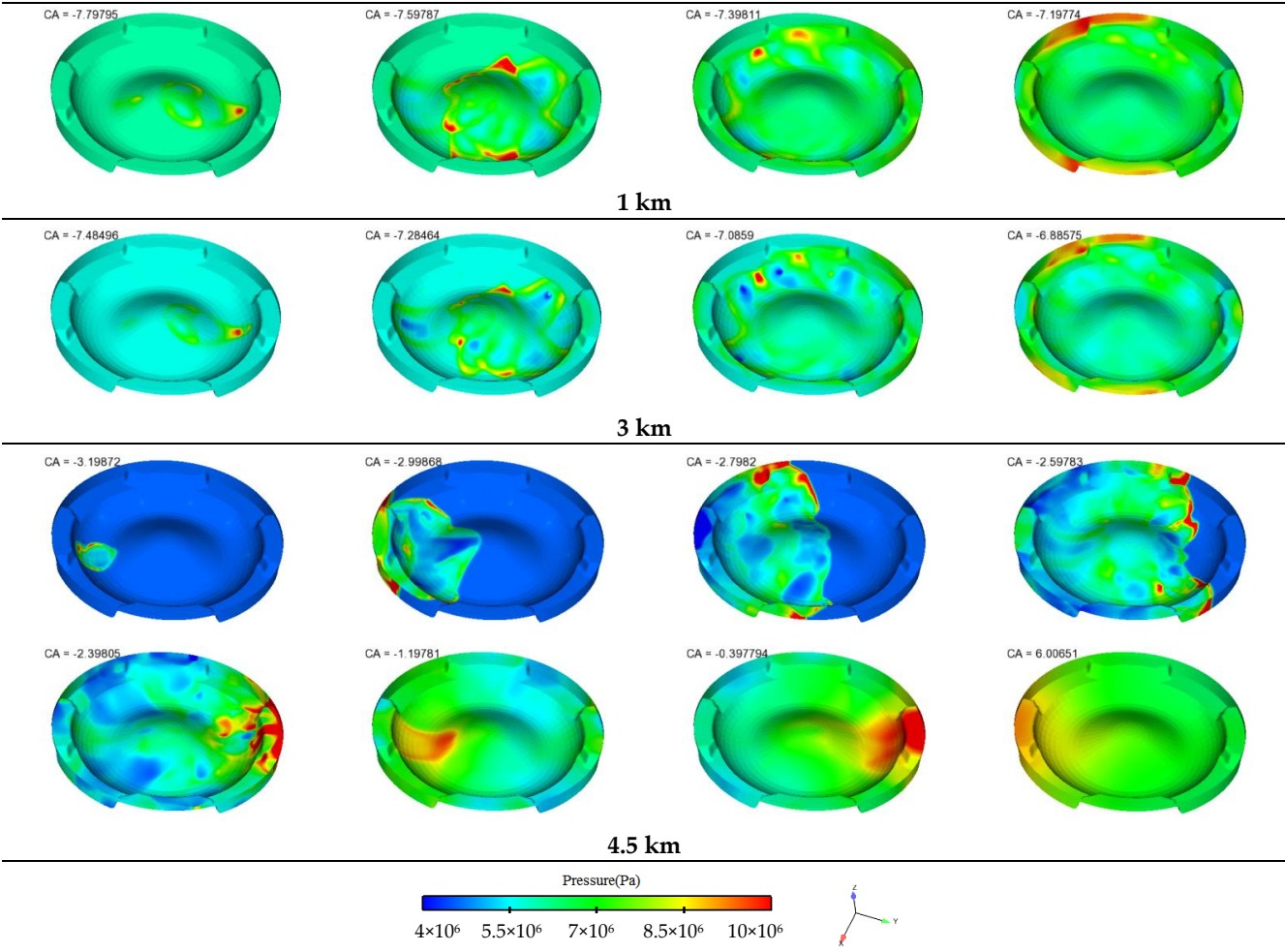

**Figure 12.** Comparison of pressure distribution at various altitudes.

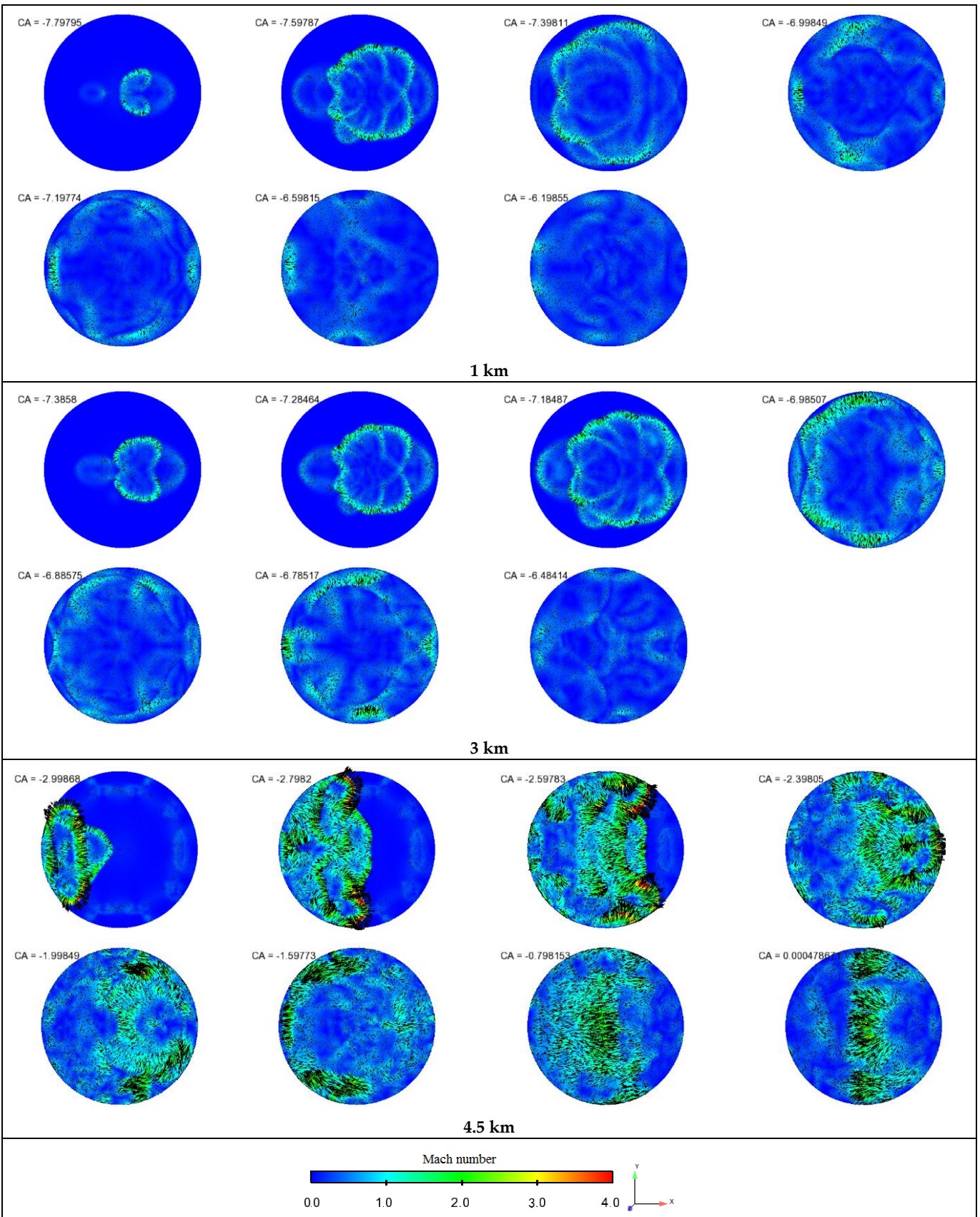

**Figure 13.** Comparison of flow velocity distribution at various altitudes.

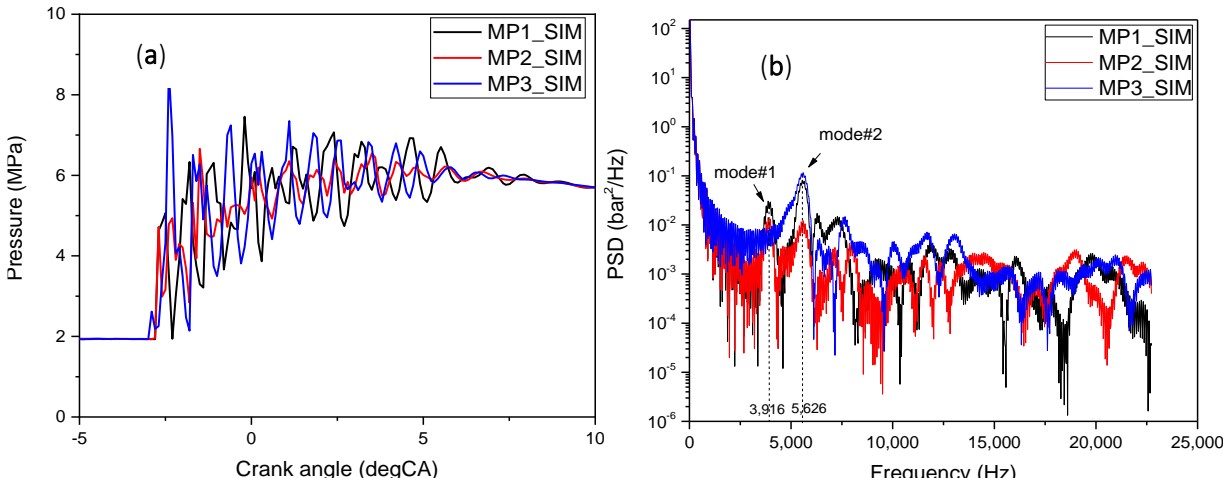

**Figure 14.** Comparisons of local pressures and PSD of monitor points at the altitude of 4.5 km: (**a**) comparison of local pressures of three monitor points; (**b**) comparison of PSD of three monitor points.

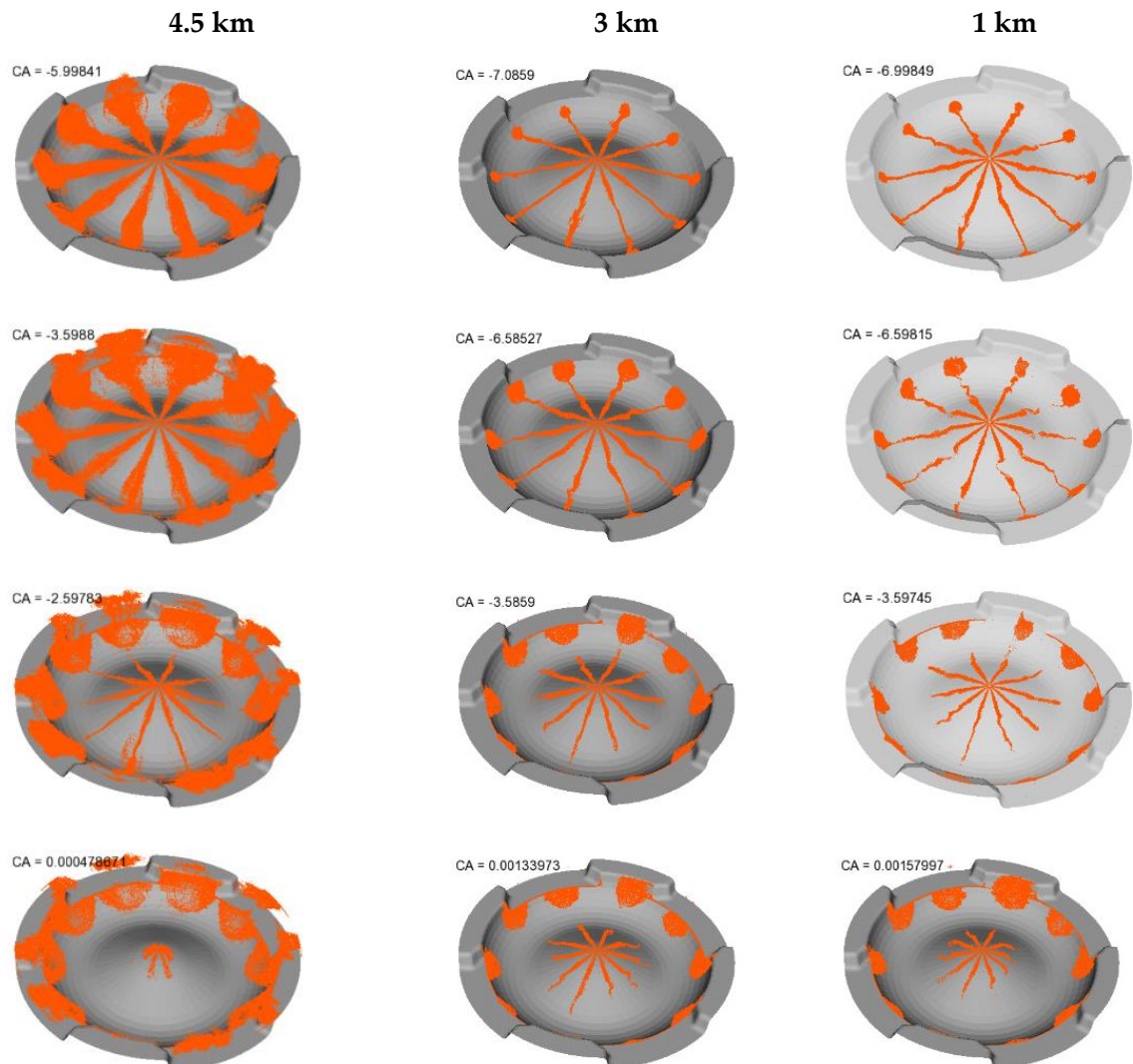

**Figure 15.** Comparison of spray distribution (liquid phase) at various altitudes.

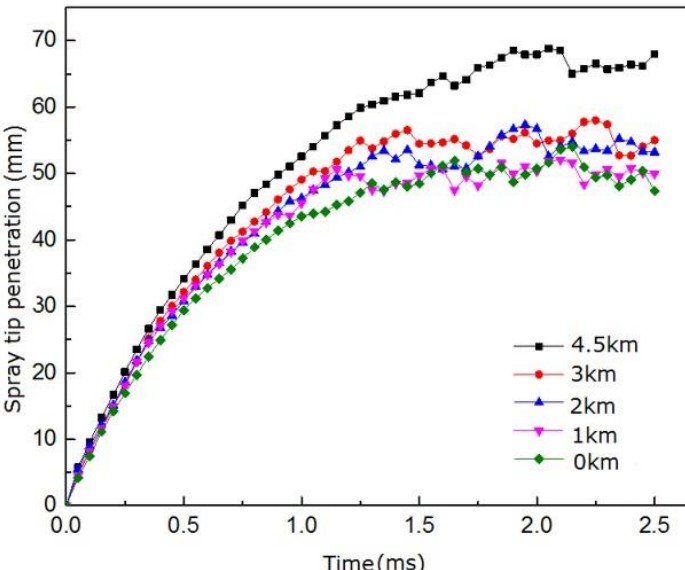

**Figure 16.** Comparison of spray tip penetration measured in a constant volume vessel at various altitudes (injection pulse = 2 ms, injection pressure = 40 MPa, and ambient temperature = 700 K).

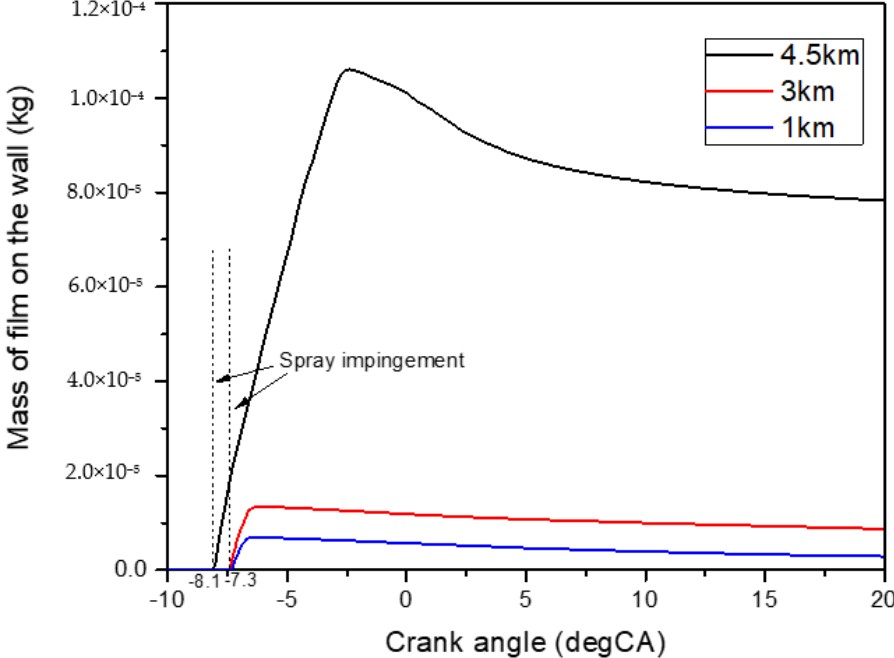

**Figure 17.** Mass of film on the wall at various altitudes.

Figure 18 shows a comparison of the ratio of fuel vapor mass to injected fuel mass at various altitudes. Firstly, the profiles of 1 and 3 km are the same except for the maximum. With the decrease in in-cylinder temperature at the same crank angle as in the increase of altitude shown in Figure 10, the evaporation rate (slope of curve) at 4.5 km was reduced compared to the cases of 1 and 3 km. Additionally, the evaporation rate of 4.5 km decreased when a considerable amount of wall film was formed. Secondly, the evaporation rate further increased with the initiation of combustion, and then the ratio decreased when the considerable fuel was burned in the cases of 1 and 3 km. On the contrary, the ratio of 4.5 km dramatically decreased without an evaporation rate increase due to the large amount of consumed fuel vapor during the early rapid premixed combustion. Finally, all the curves descended further after the end of injection (EOI).

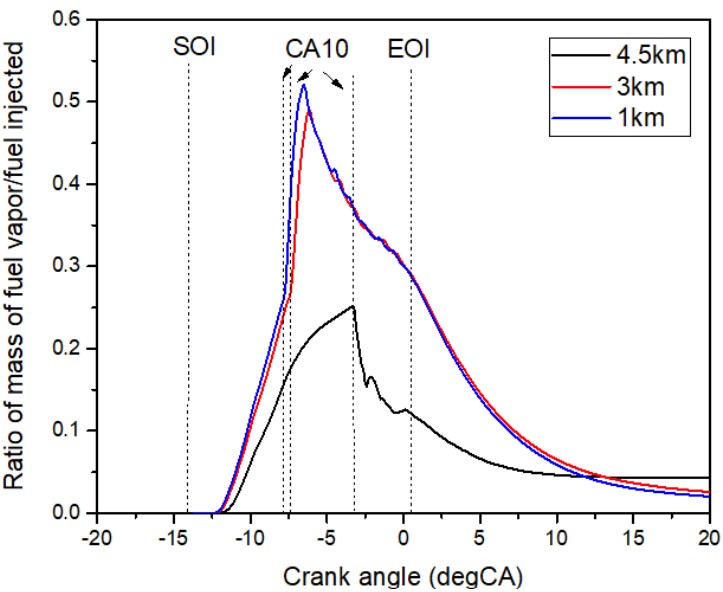

**Figure 18.** Ratio of mass of fuel vapor/injected fuel at various altitudes.

Figure 19 shows the temperature distribution at various altitudes. In the cases of altitudes of 1 and 3 km, a high-temperature zone (over 2400 K) was concentrated in the piston bowl. At the altitude of 4.5 km, a high-temperature zone occurred at −2.6 °CA ATDC, almost 4 °CA later than the other two cases. In addition to the piston bowl, an extremely high-temperature zone occurred in the squish zone around TDC. Additionally, considerable heat could be directly transferred to the piston crown without effective the cooling method since this point is far from the piston cooling channel.

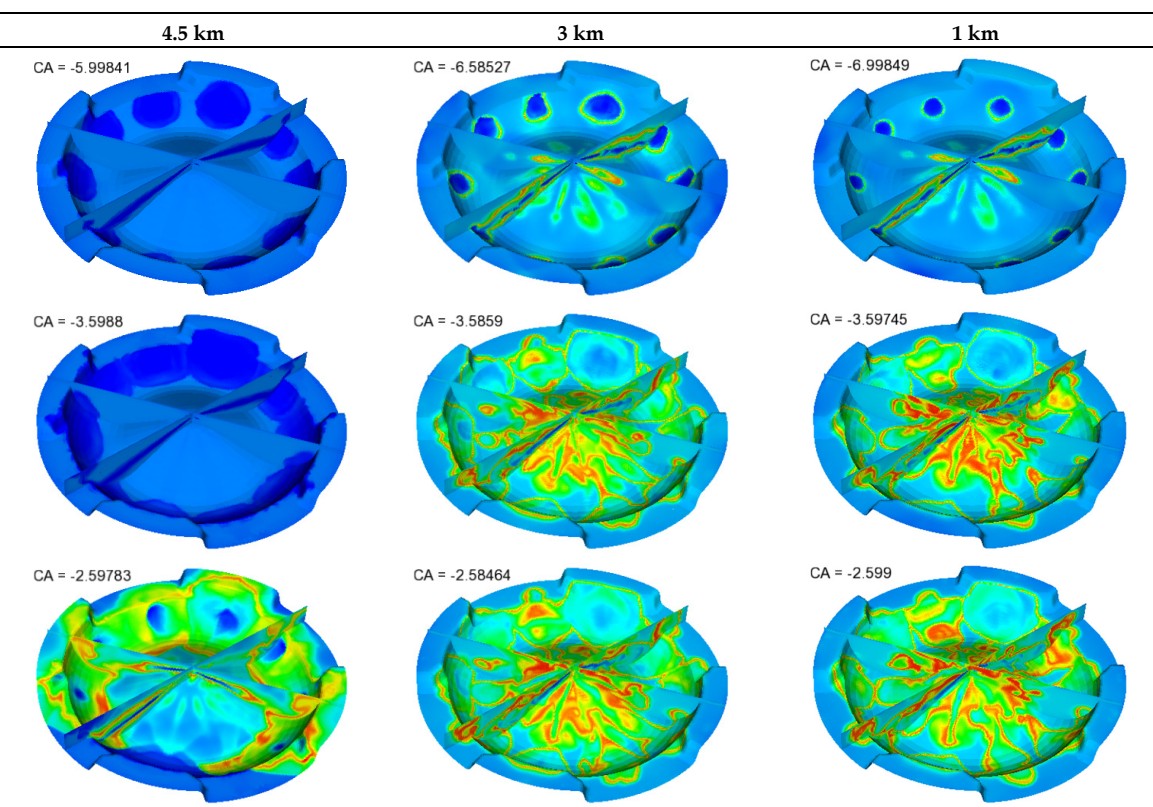

**Figure 19.** *Cont*.

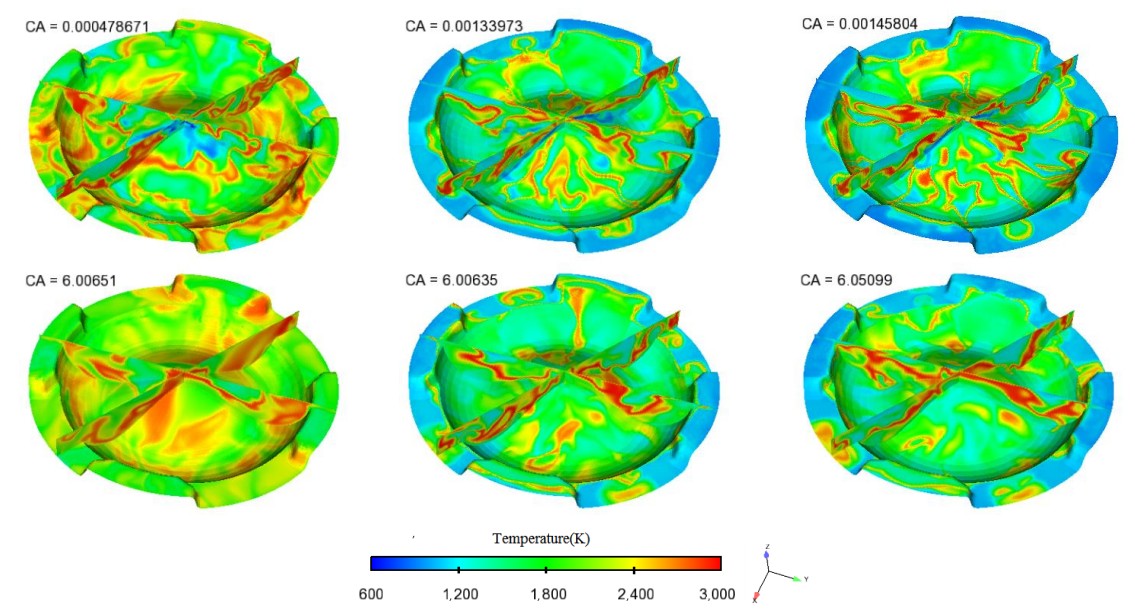

**Figure 19.** Comparison of temperature distribution at various altitudes.

Compared to the conventional diesel combustion in the cases of 3 and 1 km, the combustion style in the case of 4.5 km was similar to so-called "super-knock" SI combustion in terms of strong pressure oscillation and large pressure rise. Two main characteristics could be described as end-gas ignition and sequential combustion. Firstly, end-gas ignition requires not only the ignitable temperature and ignitable concentration of a mixture near the wall but also the thermal and concentration stratification in the whole cylinder. It is known that an ignitable mixture is firstly formed at the area near the spray front and that multiple-ignition occurs when the in-cylinder temperature reaches the ignitable temperature in normal diesel combustion. In the case of knocking combustion, decreases in compression pressure and temperature lead to worse spray atomization, worse evaporation, and severe spray impingement on the wall. After a large amount of spray impingement on the wall occurs, the ignitable mixture gradually gathers on the zone near the wall (not including the spray jet). Meanwhile, the large amount of latent heat of fuel evaporation lowers the local temperature; this was especially evident here when the temperature of the spray jet was below 600 K (as shown at the right side of Figure 20a). Thus, the ignitable zone was bordered on both the wall film and spray front, as marked with a black circle in Figure 20a, where local temperature was close to 1000 K and the local equivalence ratio was 1–1.5. Then, a deflagration was initiated (as shown in Figure 20b), and the resulting flame propagated outward. Secondly, the chemical heat release from the deflagration led to thermal expansion of the burned zone, which compressed the surrounding unburned mixture to high pressure and high temperature. Finally, sharp increases in pressure and temperature in the surrounding unburned mixture resulting from the thermal expansion of the burned zone could be observed. The deflagration flames propagated in the surrounding unburned mixture along the periphery of the cylinder. The passage travelled sequentially from this region to the center and finally arrived at the right side in a short time with supersonic speed. A reflection wave from the right-side bounced back towards the left side. Hence, pressure oscillation occurred until 6.5 °CA ATDC, as shown in Figure 14a.

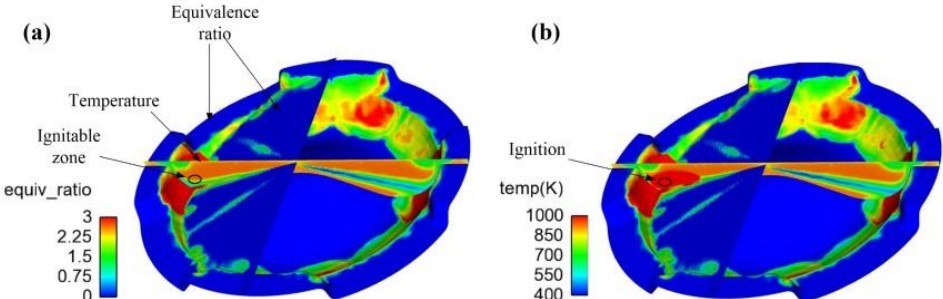

**Figure 20.** Distribution of equivalence ratio and temperature before and after ignition at 4.5 km:
(**a**) −3.4 °CA ATDC; (**b**) −3.2 °CA ATDC.

## 7. Conclusions

The effects of altitude on the main combustion characteristics—in-cylinder fluid flow, spray behavior, and pressure and temperature distribution—were analyzed with the CFD method. The major conclusions are as follows.

1. A numerical model was validated with the optical data of spray behavior and the pressure trace measured by a test bench.
2. Long ignition delay, rapid premixed and low combustion efficiency were observed under the condition of knocking combustion.
    i. The decreases in compression pressure and temperature at 4.5 km led to over 4 °CA longer ignition delays than those at 1 and 3 km.
    ii. The main combustion durations from CA10 to CA90 at 1, 3, and 4 km were 59.9, 53.5, and 15.8 °CA, respectively, and the durations from CA10 to CA50 were 14.4, 12.6, and 1.6 °CA. Thus, compared to typical diffusion combustion at 1 and 3 km, premixed combustion dominated at 4.5 km.
    iii. The combustion efficiency decreased from 90% to 47% when the combustion changed from normal combustion to knocking combustion due to severe spray impingement.
3. The processes of end-gas ignition, sequential combustion, and pressure oscillation in the knocking combustion were revealed by the numerical modeling results.
    i. A deflagration was initiated by the end-gas with the ignitable mixture near the wall due to severe spray impingement.
    ii. Instead of typical multiple-ignition, the chemical heat release from the deflagration led to the thermal expansion of the burned zone, which compressed the surrounding unburned mixture to high pressure and high temperature. The deflagration flames propagated in the surrounding unburned mixture along the periphery of the cylinder.
    iii. Due to the thermal expansion of end-gas ignition, the pressure wave propagated from the one side to the other side of the cylinder within 0.8 °CA, and the mean propagation velocity of pressure wave was 1364 m/s. A typical reciprocating pressure oscillation was observed.

In the future, more research on strategies for the diesel knock control will be conducted. Additionally, more combustion system design work is required to attenuate knock.

**Author Contributions:** Conceptualization, H.L.; Data curation, C.L.; Formal analysis, W.Z.; Methodology, F.L.; Project administration, H.L.; Resources, Y.L. (Yaozong Li); Software, X.Z.; Supervision, Y.L. (Yufeng Li); Validation, R.C.; Writing—original draft, H.L. All authors have read and agreed to the published version of the manuscript.

**Funding:** This research was funded by National Natural Science Foundation of China grant number 51476151 and Tianjin Natural Science Foundation grant number 17JCQNJC06700.

**Institutional Review Board Statement:** Not applicable.

**Informed Consent Statement:** Not applicable.

**Data Availability Statement:** Not applicable.

**Conflicts of Interest:** No conflict of interest exits in the submission of this manuscript, and manuscript is approved by all authors for publication. I would like to declare on behalf of my co-authors that the work described was original research that has not been published previously, and not under consideration for publication elsewhere, in whole or in part. All the authors listed have approved the manuscript that is enclosed.

## Abbreviations

The following abbreviations are used in this manuscript:

| | |
|---|---|
| AMR | Adaptive Mesh Refinement |
| ATDC | After Top Dead Center |
| BTE | Brake Thermal Efficiency |
| CA | Crank Angle |
| CFD | Computation Fluid Dynamics |
| CI | Compression-Ignition |
| CNERI | China North Engine Research Institute |
| COV | Cycle-to-Cycle Variation |
| IM EP | Indicated Mean Effective Pressure |
| KI | Knock Intensity |
| LES | Large Eddy Simulation |
| MP | Monitor Point |
| PPRR | Peak Pressure Rise Rate |
| PSD | Power Spectral Density |
| SI | Spark-Ignition |

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
