# Peer review of "Numerical Study of Knocking Combustion in a Heavy-Duty Engine under Plateau Conditions"

_energies, doi:10.3390/en15093083_

Round 1

Reviewer 1 Report

In the reviewed paper, a numerical analysis of the operation of a diesel engine under high altitude conditions was conducted. Under these conditions, the oxygen content and air density differ significantly from conditions considered normal.

The CFD method was used to perform the analysis effects of altitude on the main combustion characteristics, in-cylinder fluid flow, spray behaviour and pressure and temperature distribution. Three-dimensional fluid flow and combustion modelling with turbulence, evaporation, droplet breakup, spray-wall interaction, particle interaction, collision and combustion submodels is described in detail in section 2

The problem under consideration has a large utilitarian dimension and a somewhat smaller scientific one. Nevertheless, the article is worth publishing after supplementation concerning the conditions of implementation of experimental research.

The reviewed paper does not provide precise information on the experimental research methods used and the research apparatus used. The potential reader is not able to find out with what uncertainty (error) the test results were recorded and whether the comparison of experimental results with the results of numerical simulations presented in the article refers to a single test or to the average value for a specified number of tests. Therefore, Section 3 should include supplemental data on experimental studies. It would also be interesting to answer the following questions.

  1. Why do Figures 4 and 5 compare experimental and numerical images of spray behaviour for different ambient temperatures 300K and 600K?
  2. What would the graph in Figure 5 look like for these experimental conditions but with ambient temperature of 300K?
  3. With what accuracy does the simulation study replicate the experiment?

An evaluation of the conclusions can be made after the authors have completed the data on the accuracy of the research methods used.

Author Response

Thank you for your kind review. The relevant information of experimental research has been added in the Section 2 of Experimental Methods. The other detailed revision is described as follows:

Q: 1.Why do Figures 4 and 5 compare experimental and numerical images of spray behaviour for different ambient temperatures 300K and 600K?

A: The different temperatures correspond to the statuses of non-vapored and vapor of spray. In order to show the calibration of spray breakup model, the comparison of experimental and numerical spray behavior was made.

Q: 2.What would the graph in Figure 5 look like for these experimental conditions but with ambient temperature of 300K?

A: The experimental vapor spray penetration was measured under the ambient temperature of 600K, not the condition of 300K. As shown in Figure 4, the penetration at 2ms is about 50mm, and the value in Figure 5 is over 60mm.

Q: 3.With what accuracy does the simulation study replicate the experiment?

A: In terms of spray behavior, as shown in Figure 4, the shape and penetration of spray calculated by CFD basically agrees with the high-speed photos.  Also the vapor penetration length predicted achieves a good agreement with measurement under high temperature of 600K, as shown in Figure 5. As respect to pressure trace, as shown in Figure 6, there is a noticeable difference between the numerical ensemble-averaged pressure and measurement is observed at the altitude of 4.5km. It is caused by the pressure inhomogeneity resulting from strong pressure oscillation. However, the local pressure predicted by CFD in the cell of the location near the pressure sensor achieves a good agreement with measurement. Also, a good consistency can be found in the cases of altitudes of 3km and 1km, and the differences between experimental and numerical results are basically within the error bar of 2%, as shown in Figure 7 and Figure 8.  

Please find the updated manuscript attached.

Reviewer 2 Report

Numerical study of knocking combustion in a heavy-duty engine under plateau conditions

This paper used CFD method to analyze the effects of altitude on the main combustion characteristics. Comparisons of the results were made with a bench test from the same CFD platform CONVERGE. The study found out that the decrease of compression pressure and temperature at 4.5km leads to over 4o CA longer of ignition delay than those of 1km and 3km. The combustion efficiency decreases from 90% to 47% when the combustion changing from normal combustion to knocking combustion due to the severe spray impingement. The innovation of the study is low.  There are some difficulties in appreciating the results from the comparisons to show the better performance of some improved technologies.

  1. Introduction.
  • Please state the brief introduction of the current research in the Section of Introduction. The main innovation of the paper should be clarified.
  1. Numerical Model Setup
  • Before this section, an additional section of the basic theory should be included in the paper.
  • The control volume of the mesh should be specified with a certain number not with a range. In paper line 71.
  • The used numerical algorithm should be stated in the paper.
  • The geometry of the adopted model should be shown.
  • Figure 2 should be well shown.
  1. Validation of Numerical Modelling
  • The quality of Figure 13 should be improved.
  • Postprocessing of the pressure should be made in Figure 14 (a). MP1_SIM cannot be seem, also in the area of crank angle from 0 to 10, the data cannot be seen.

Author Response

Thank you for your kind review. The detailed revision is described as follows:

Q:  1.Please state the brief introduction of the current research in the Section of Introduction. The main innovation of the paper should be clarified. 

A:  The current research of bench test has been introduced in the lines (38-43) and the main innovation of this study has been highlighted in the last paragraph of Section 1 of lines (70-76) from the terms of effect of altitude on the combustion characteristics and the disclosure of the ignition beginning and combustion propagation for CI knocking combustion.

Q:  2. Before the section of Numerical Model Setup, an additional section of the basic theory should be included in the paper. The control volume of the mesh should be specified with a certain number not with a range. In paper line 71. The used numerical algorithm should be stated in the paper. The geometry of the adopted model should be shown. Figure 2 should be well shown.

A:  Thank you for your good advice. A section of Numerical Methodology has been added in order to introduce the related algorithm and model theory. And the specified cell number of computational mesh at TDC and maximum of mesh number have been stated in the Section 4.

Q:  3. The quality of Figure 13 should be improved. Postprocessing of the pressure should be made in Figure 14 (a). MP1_SIM cannot be seen, also in the area of crank angle from 0 to 10, the data cannot be seen.

A: The Figure 13 has been updated with a higher resolution. And the range of Figure 14 (a) has been modified in order to identify the lines of individual monitor points.

Please find the updated manuscript attached.

Reviewer 3 Report

This work investigates the knocking combustion of a CI engine under at different altitudes with the numerical simulation. Effects of altitude on the pressure oscillation, temperature, heat release and ignition delay of CI combustion were analyzed numerically. The results obtained in this study can be useful for the understanding of the knocking combustion development in the CI engine under high altitude conditions. However, this paper can be accepted for the publication after the following questions are solved.

  1. The objective of this study and its contribution to the related fields should be highlighted.
  2. More details about the numerical model should be provided, such as domain size, grid size and discretization methods.
  3. line 108: “As shown in Figure 10, maximum combustion temperature increases from 2090 K at 1km to 2350 K at 4.5km”. According to the results, the heat release and pressure are both decreased at 4.5km. Why is the peak combustion temperature increased considerably? More explanation about this should be added.
  4. line 116, why does the rapid premixed process occur at 4.5 km?
  5. Fig 12, 13 and 19 are suggested to be replaced by the high-resolution images.
  6. How to obtain the Conclusion 2. ii. (line 209-210)?
  7. Instead of the present conclusion 3.(line 213-214), the significant variations of end gas ignition, sequential combustion and pressure oscillation in the knocking combustion should be concluded specifically.
  8. The grammatical mistakes should be checked thoroughly.

Author Response

Thank you for your kind review. The detailed revision is described as follows:

Q:  1. The objective of this study and its contribution to the related fields should be highlighted.

A: Thank you for your great advice. The objective of this study has been highlighted in the last paragraph of Section 1 from the terms of effect of altitude on the combustion characteristics and the disclosure of the ignition beginning and combustion propagation for CI knocking combustion.

Q:  2.More details about the numerical model should be provided, such as domain size, grid size and discretization methods.

A:  The related information has been added in the updated Figure 2 and the first paragraph of Section 4.

Q:  3.line 108: “As shown in Figure 10, maximum combustion temperature increases from 2090 K at 1km to 2350 K at 4.5km”. According to the results, the heat release and pressure are both decreased at 4.5km. Why is the peak combustion temperature increased considerably? More explanation about this should be added.

A:  Although the maximum pressure and the integrated heat release decreases at 1km to 4.5km, the maximum of heat release rate increases by 57% as listed in Table 3. This indicates the occurrence of rapid and short heat release at 4.5km. Thus the maximum combustion temperature increases.

Q:  4.line 116, why does the rapid premixed process occur at 4.5 km?

A: In the case of 4.5km, the main combustion duration from CA10 to CA90 is 15.8 o CA. And the duration from CA10 to CA50 is only 1.6o CA as listed in Table 3, indicating a rapid premixed combustion occurs.

Q:  5.Fig 12, 13 and 19 are suggested to be replaced by the high-resolution images.

A:  According to your kind advice, the related graphs have been updated with high-resolution ones.

Q:  6.How to obtain the Conclusion 2. ii. (line 209-210)?

A:  That is a good question. The main combustion durations from CA10 to CA90 at 1km, 3km and 4km are 59.9 o CA, 53.5 o CA and 15.8 o CA, individually. And the durations from CA10 to CA50 are 14.4 o CA, 12.6 o CA and 1.6o CA. Thus, compared with typical diffusion combustion of 1km and 3km, premixed combustion was dominated at 4.5km.

Q:  7.Instead of the present conclusion 3.(line 213-214), the significant variations of end gas ignition, sequential combustion and pressure oscillation in the knocking combustion should be concluded specifically.

A: Thank you for your great advice. The third conclusion has been expanded into the specified part. Please find the lines (307-314).

  1. Adeflagration is initiated by the end gas with ignitable mixture near the wall due to the severe spray impingement.
  2. Instead of typical multiple-ignition, the chemical heat releasefrom the deflagration leads to thermal expansion of the burned zone, which compresses the surrounding unburned mixture to high pressure and high temperature. The deflagration flames propagate in the surrounding unburned mixture along the periphery of the cylinder.
  • Due to thermal expansion of end-gas ignition, the pressurewave propagates from the one side to the other side of cylinder within 0.8 o CA and the mean propagation velocity of pressure wave attains 1364 m/s. A typical reciprocating pressure oscillation is found.

Q:  8.The grammatical mistakes should be checked thoroughly.

A:  The related grammatical mistakes have been corrected.

Please find the updated manuscript attached.

Round 2

Reviewer 1 Report

Thank you for sending the clarification of my concerns formulated in the review questions. I accept the new revised version of the paper.

Reviewer 2 Report

The authors have changed accordingly. The present form can be accepted.

Reviewer 3 Report

The paper has been revised properly according to the comments. Now it is ready for publication.